# Using Classical and Operant Conditioning to Train a Shifting Behavior in Juvenile False Water Cobras (*Hydrodynastes gigas*)

**DOI:** 10.3390/ani12101229

**Published:** 2022-05-10

**Authors:** Michelle L. Williams, Lori A. Torrini, E. Joseph Nolan, Zachary J. Loughman

**Affiliations:** 1Department of Organismal Biology, Ecology, and Zoo Sciences, West Liberty University, West Liberty, WV 26074, USA; enolan@westliberty.edu (E.J.N.); zloughman@westliberty.edu (Z.J.L.); 2Behavior Education at Spirit Keeper Animal Sanctuary, Handle Road, Yoder, CO 30315, USA; behavioreducationllc@gmail.com

**Keywords:** snake training, operant conditioning, positive reinforcement, reptiles, enrichment

## Abstract

**Simple Summary:**

Positive reinforcement training, a form of operant conditioning that rewards learners for specific behaviors to increase the frequency of that behavior, is widely used as a management tool in modern zoological facilities. This type of training offers several benefits, including improved safety, reduction in animal stress, increased choice and control, and many others. Snakes have been shown to learn using operant conditioning, yet the use of positive reinforcement training is not a widespread practice for these animals. A shaping plan, which describes the steps necessary to achieve the ultimate behavioral goal of following a target into a secondary container from their primary enclosure, was developed for false water cobras (*Hydrodynastes gigas*). Snakes were given food rewards for performing behaviors related to each goal detailed in the plan. Completion of this shaping plan resulted in a reduction of stress behaviors as the training progressed, as well as a decrease in time between the presentation and snakes interacting with the target.

**Abstract:**

All animals have the capacity to learn through operant conditioning and other types of learning, and as a result, zoos and other animal care facilities have shifted towards the use of positive reinforcement training to shape the behavior of animals under their care. Training offers animals the choice to participate in their own husbandry routines and veterinary procedures, while also providing mental stimulation. By adopting these practices, the welfare of animals in human care has improved, but it has not been applied equally across taxa. Snakes are frequently overlooked in the discussion of choice and control in a captive setting, likely due to the historical misinterpretation of their intelligence and behavioral needs. In this study, a shaping plan was developed for 28 juvenile false water cobras (*Hydrodynastes gigas),* a rear-fanged venomous species, from four clutches. Snakes were rewarded with food when completing behaviors related to the ultimate goal of following a target into a shift container. The purpose of this study is to incorporate the trained behaviors in routine husbandry practices, while preventing unnecessary stress in the snakes and risk to the keeper.

## 1. Introduction

Progression in zoo animal welfare research has been a driving force behind the increase in positive reinforcement training used to modify behavior in animals under human care in modern zoos. By adopting this methodology, animals under captive management are taught to voluntarily participate in their own care, including veterinary and routine husbandry procedures. Positive reinforcement is a form of operant conditioning, where a reinforcing stimulus is added to the training environment following a behavior to increase future frequency of the behavior. The benefits of using positive reinforcement techniques are well documented. The welfare benefits of cooperative care training using positive reinforcement include stress reduction, increased opportunities for physical exercise, mental stimulation, and improved trainer/animal relationships [1]. Target training, or targeting, is when an animal learns to touch an object (target) with part of their body or follow a target to earn reinforcement. Targeting is commonly used in animal training and is a foundational skill used as a starting point for other behaviors, such as stationing or shifting. Stationing refers to teaching an animal to go a specific area and wait, or to position on a specific object. Shifting refers to moving an animal from one place to another [2].

All animals are capable of learning. Animals in their natural habitats receive reinforcement through various stimuli encountered because of their actions; both classical and operant conditioning have been documented in wild animals [3]. Consequently, most animal species can be trained in human care. In zoos, positive reinforcement techniques have been successfully implemented with mammals and have been documented in non-mammals, particularly birds and aquatic animals. Savastano et al. [4] described a massive effort of developing a less invasive training program for 17 species of New World primates at the Bronx Zoo, where a variety of behaviors, including hand and syringe feeding, targeting, and crate training, were taught. Savastano et al. [4] reported a reduction in aggressive behaviors and more visibly comfortable animals because of the training program.

Callealta et al. [5] reported the use of positive reinforcement training to allow for frequent blood sampling and vaginal swabbing of African lions (*Panthera leo*), veterinary procedures that typically require physical or chemical restraint in untrained animals. By training these behaviors, keepers obtained valuable physiological information while minimizing stress. Mattison [6] reported training birds and small mammals for a variety of husbandry behaviors at the Point Defiance Zoo and Aquarium. Trained behaviors were divided into three levels, foundation, intermediate, and advanced, based on difficulty. Foundation behaviors included crate and scale training, while intermediate behaviors included trimming and allowing restraint. Accepting injections or masking anesthesia were examples of advanced behaviors.

Fish and aquatic invertebrates can be trained for husbandry and veterinary procedures, as documented by aquarists in recent years [7]. Using classical and operant conditioning techniques, eagle rays (*Aetobatus narinari*) at Disney’s Epcot The Living Seas were trained to station on a target for feeding, in addition to transferring between divers [8]. Training of husbandry behaviors, including targeting, allowing tactile contact, voluntary blood draws, and voluntary weighing, has been reported for zebra sharks (*Stegostoma fasciatum*), allowing these animals to choose to participate in most of their care [9]. Moffatt [10] described training a giant Pacific octopus (*Enteroctopus dofleini*) to come to the surface of the water when the water was slapped and to climb into a container for weighing. These examples note improvements in animal welfare and health, further supporting that training is an important husbandry management tool across taxa and that providing choice is conducive to welfare improvements.

The cognitive abilities of reptiles and amphibians have traditionally been overlooked resulting in an absence of documented training and enrichment protocols [11]. Compared to the body of literature that exists for training mammals and birds, there are relatively few studies assessing the learning capabilities of reptiles. Papers published on reptile learning demonstrate that reptiles do benefit from the use of positive reinforcement techniques and can learn through operant conditioning. For example, Weiss and Wilson [12] trained four Aldabra tortoises *(Aldabrachelys gigantea*) to hold at a target and allow for blood to be drawn from their necks. Physical restraint, while not usually producing physical harm, induces stress in animals and people [13]. Blood drawing procedures can be stressful, especially if restraint is involved, and the use of positive reinforcement with the Aldabra tortoises minimized the amount of stress the tortoises experienced. In another study, Gutnick, Weissenbacher, and Kuba [14] found that Aldabra tortoises, as well as Galapagos giant tortoises (*Chelonoides nigra*), learned a targeting behavior faster when trained with conspecifics and could differentiate between their specific target and an unconditioned one. Positive reinforcement training has also shown to be beneficial when working with potentially dangerous reptiles, such as large crocodilians. Augustine and Baumer [15] described training a Nile crocodile (*Crocodylus niloticus*) to target, station, and hold for blood draws, a procedure that would otherwise require capture and restraint.

Training literature documenting learning in snakes, while limited, does exist and includes studies focused on operant conditioning. Kleinginna [16] found that indigo snakes (*Drymarchon couperi*) can be conditioned to press a key to obtain water. In another study, wild Burmese pythons (*Python bivitattus*) were first conditioned to consume multiple small prey items and then were taught to press a button, but only when the button was illuminated, to open a door and receive a food reward [17]. These studies describe a form of targeting using buttons as visual cues.

Snakes are animals that strongly rely on chemical senses to obtain environmental information [18], but snakes can also respond to other types of cues and learn to follow a scent trail to receive a food reward, as demonstrated by Begun et al. [19]. Based on these and other studies, snakes are capable of learning through operant conditioning. It is plausible that snakes can benefit from the positive outcomes associated with choice and control granted through positive reinforcement training, which are often touted for other taxa. However, Hellmuth et al. [2] cites only three species of snake, the false water cobra (*Hydrodynastes gigas*), king cobra (*Ophiophagus hannah*), and green mamba (*Dendroaspis angusticeps*), in their examples of zoo reptile training.

Professional animal trainers have documented multiple methods as effective, successful, and enriching for training snakes and other reptiles. Kish [20] reports training husbandry behaviors in reptiles, including targeting, stationing, shifting, and voluntary handling. To initiate a training protocol, the author first identifies natural history information, as well as relevant behavioral data, such as how the snake uses their space and how different types of interactions affect the presence of relaxed behaviors or stress signals. Kish [20] describes frequent use of novel objects, particularly food puzzles of varying complexity, to encourage problem solving and natural hunting behaviors. Amelia [21] expands on the use of puzzle feeding for snakes, describing an increase in species-specific behaviors and more relaxed behavior around food as two benefits of this form of enrichment. Torrini [22] explains two methods for training snakes to voluntarily shift out of their primary enclosure. In the first method, operant conditioning is used to teach snakes that the presentation of a specific visual cue indicates that they can move to a different location to receive a food reward. For this method, Torrini [22] explains that the shifting behavior can be trained by reinforcing the snake when it interacts with a specific container on its own, by luring the snake into the container, or by physically placing snakes—which are comfortable and relaxed with handling—into the container for feeding. The final behavior is for the snake to move into the shift container or to the station autonomously, when it is presented. In the second method, target training is used, and once the snake is consistently interacting with the target, the target can be moved short distances and eventually into a container. All three authors express that monitoring behavior and body language for signs of fear, anxiety, and distress is of utmost importance to ensure that training sessions and enrichment are positive experiences for learners.

In this study, false water cobras (*H. gigas*), were trained to follow a target out of their primary enclosure and into a shift container. *Hydrodynastes gigas* are active predators with strong visual perception [23], that eat frequently in the wild—compared to other species of snakes—and have a diverse diet [23]. The aforementioned characteristics all serve as a justification for the choice to use this species, as application of positive reinforcement techniques will make it significantly easier to manage adults that were trained as hatchlings. As a direct effect of feeding frequency, these snakes require routine cleaning of their enclosures; *H. gigas* are also fast-moving and grow quickly to large sizes [23]. While their venom has not been documented as medically significant, it is possible for humans to have allergic reactions to it, and their large adult size can pose a risk to human safety if an individual were to bite. As hatchlings, they are easily stressed and display dramatic behaviors, such as hooding, tail-whipping, and feigning death in response to stressors. Training of this species has been recorded before, with keepers using targeting and shifting behaviors trained using a scented target to move individual snakes out of a shared enclosure for feedings; it was noted that this procedure increased keeper safety and reduced handling time of the snakes [24]. The primary aim of the present study was to develop an effective plan for target training and shifting for *H. gigas* to increase safety and reduce management stress for animals and keepers.

## 2. Materials and Methods

### 2.1. Subjects

A total of 28 hatchling *H. gigas* from four clutches were initially selected for this study, with 21 individuals having prior training experience using an aversive and invasive method. Each clutch was incubated in separate containers at 27.8 °C (82 °F), and all individuals hatched during summer 2020. Seven individuals were randomly chosen from each clutch (Table 1). The oldest and youngest clutches had the same set of parents, but the other two clutches had unique parent pairings. After hatching, all subjects were housed initially in shoebox size tubs in a temperature-controlled rack system with a hide, water dish large enough for soaking, and substrate mixture of cypress mulch, topsoil, sand, and coco coir (Figure 1). The warm side of their enclosures was maintained at 31.1 °C (88 °F) before and during the study. Once snakes had reached the approximate length of their initial tub, they were transitioned into larger enclosures by mixing their old substrate with fresh substrate. Furnishings for the larger enclosures were similar to initial housing conditions, but an additional hide and larger water bowl were added. All care and daily maintenance—including daily spot cleaning, feedings, and water changes as needed—was performed by the trainer. Snakes were offered appropriately sized rodents twice a week before and during training sessions. Occasionally, quail were offered as reinforcement. During the training sessions, individuals were offered up to four prey items adding up to the weight of the largest prey item they could consume.

### 2.2. Shaping Plan Development

Using a template created by Torrini [25], a shaping plan was made to detail the behavioral approximations necessary to achieve the final shifting behavior. Appendix A displays a summary of the training routine the snakes followed during the study. Several aspects of this species’ natural history were researched prior to development of the plan, including: diet, feeding frequency, and which colors they are likely to perceive. Information about diet and feeding frequency were used to determine the training schedule, and a target color was chosen based upon which colors they could likely perceive. It was determined that this species eats frequently in the wild, and that snakes are capable of perceiving blue and green [26,27] This information was used to justify the frequency of feedings and the colors of the targets used. Individual history was also taken into consideration, as each clutch had individuals displaying different behaviors that required modifications to the shaping plan. While the original shaping plan (Appendix A) was attempted for each individual in the study, if a snake consistently refused to participate in the training, modifications were made to reduce unnecessary stress. These modifications are discussed further in the following section. Observations of snake activity were carried out randomly prior to the beginning of training to determine which time of day the snakes were most active.

### 2.3. Positive Reinforcement Training

For clutches #1–3, training sessions were initiated from the beginning of November 2020 and occurred twice weekly until the end of January 2021. Training sessions for clutch #4 also began in November 2020 and occurred at the same frequency but continued until the end of May 2021. All training sessions for each group were initiated between the hours of 2:00 p.m. and 4:00 p.m., as previous observations indicated this was when they were most active. Each session was conducted by the same trainer. A blue ball target on a stick was used to train the youngest clutch, while a novel target, a green ball on a stick, was used for the clutches that had previous aversive experiences when trained with a blue ball target. Learners had opportunities to earn up to four food rewards per session. Snakes were not trained if they were in shed. Classical conditioning was initially used to indicate that the presence of the target meant the snake would receive food, by pairing the target with food. In early sessions, the target was gradually introduced to the snakes by presenting the target a distance away from the snake and gradually moving the target closer to the learner in later sessions; this was carried out to avoid fearful behaviors in response to the target.

The snakes were given the opportunity to choose if they wanted to participate in each session or not. If snakes displayed any behaviors indicative of severe stress, such as full body flattening or retreating into their hide, they were exempted from training for that session. Following a severe stress behavior, if the snake remained outside of its hide and appeared interested, the session continued; if the snake continued to hide or display other severe stress behaviors, the session ended. During early sessions, if snakes were hiding during the scheduled training times, they were not trained. For every training session, individual learners, with some exceptions, were brought to a contained training area.

Initially, this involved setting their tubs inside a larger tub from which they could not escape (Figure 2). As the snakes grew and their primary enclosure no longer fit within the initial training environment, this containment area was upgraded to a small tent (Figure 3). As the snakes progressed through their training, novel items, such as glove boxes, deli cups, and the shift tub were introduced to the training environment. Snakes were given time to explore the containment areas and interact with novel items outside of training sessions.

When the shift tub was first presented, it contained a substrate mixture of coco coir and sand, as well as a water bowl. This was carried out to make this environment more attractive to the snakes, encouraging them to voluntarily enter the shift tub. Once snakes were confidently following the target into the shift tub, these furnishings were removed. As snakes grew, a larger shifting tub was introduced, initially with the same substrate mixture as the previous shift and a large water tub inside. The same process for presenting the smaller shift tub was repeated for the larger one.

Some modifications had to be implemented for certain individuals. One individual from clutch #4 resisted taking food from tongs, so measures were taken to help the learner adjust to this. This involved setting the tongs near the food item until the snake was comfortable enough to approach the food item in the presence of the tongs. Three individuals from the clutch #1 who had aversive experiences during past training sessions continued to display fearful behaviors towards the new ball target. Consequently, a different target—a blue lid—was used for two of them, while a stand-alone green ball was used for the remaining individual. These individuals were trained within their tubs in their rack, rather than taken to a training area like the others. Data was not recorded for these individuals, but they still participated in the training. From clutch #3, one individual refused food regardless of how it was presented, so they were removed from the study. One individual from clutch #4 was housed in a glass aquarium after showing exemplary learning during the targeting process; however, this snake was removed from the study due to husbandry issues that caused inconsistent feeding.

Individual learning records detailing behavior and progress during training sessions were created for each snake. Sessions were recorded via video and/or notes taken during training. Videos were assessed for the frequency of stress behaviors, such as moving away from the target, hiding their heads, and exiting the training environment. A species-specific ethogram, displayed in Table 2, was created using a combination of observations from previous interactions and a snake specific behavior chart created by Torrini [28]. Stress behaviors were labelled as either yellow zone behaviors for signs of moderate stress or red zone behaviors for signs of severe stress. To identify the rate of learning, response latency, the amount of time between the presentation of the target and the desired response, was calculated for each session that video recordings were available for.

### 2.4. Data Analysis

Data were analyzed using R version 4.1.3 [29]. Due to the difference in learning history and the length of training procedures between the fourth clutch and the older three clutches, statistical analyses were conducted on the fourth clutch individually, while the measurements for the three oldest clutches were combined. Linear regression was performed to assess the progression of response latency given the number of training sessions, for these two groups. Subsequently, the ratio of observed stress behaviors to the quantity of time monitored was calculated for use as the response variable in linear regression modeling. This was performed separately for moderate and severe stress behaviors, given the number of training sessions as a predictor in both. For incomplete repetitions, where the snake did not complete the task at hand, the period of time monitored was fixed at 60 s.

## 3. Results

### 3.1. Response Latency

Over a total of eight training sessions, the oldest three clutches had an average response latency of 38.75 s. Targeting to the shift containing was the furthest approximation that this group of snakes completed. The response latency for these clutches showed a slight increase as the training progressed (Figure 4). However, the linear regression for their response latency showed that this increase was neither significant, nor fit the data well (r^2^ = 0.0003, *p* = 0.31). Clutch #4 participated in a total of 51 training sessions and had an average response latency of 43.8 s. This clutch performed all the behaviors described in the shaping plan, including the full behavior of shifting in and out of their primary enclosures. A decrease in response latency over the 51 sessions was observed for clutch #4 (Figure 4). The linear regression for this clutch indicated there was a significant decrease in response latency, although model goodness of fit to the data was poor (r^2^ = 0.08, *p* < 0.001).

### 3.2. Stress Behaviors

There appeared to be a slight decrease in yellow zone behaviors over number of training sessions for the oldest three clutches (Figure 5), though it was not significant (r^2^ = 0.008, *p* = 0.24). Interestingly, for clutch #4, this measurement increased over the progression of the training sessions (Figure 5). However, this increase was not significant (r^2^ = 0.01, *p* = 0.07). Across all clutches, hooding was the most common yellow zone behavior displayed. Both the oldest three clutches and clutch #4 displayed decreases in red zone behaviors over the number of training sessions (Figure 6). For the oldest three clutches, this change was not significant (r^2^ = 0.02, *p* = 0.14). In contrast, the decrease in this measurement for clutch #4 was significant (r^2^ = 0.14, *p* < 0.001). With the exception of the youngest and oldest clutch, each clutch had a different modal red zone behavior. Clutches #1 and #4 both hid when experiencing severe stress, while clutch #3 displayed body flattening most often. In response to severe stress, the most common red zone behavior performed by clutch #2 was an attempt to bite the trainer; this was the only clutch to do this behavior.

### 3.3. Training Step Completion

For clutch #4, the number of training sessions it took for each snake to perform the subsequent training step, starting with classical conditioning and ending with returning to their enclosure, is displayed in Table 3. For the first step in the training plan, classical conditioning, this group required an average of 9.2 sessions (SD = 1.32, range 7–11) before moving to the following step. Targeting in and out of their enclosure appeared to be learned more quickly than the other behaviors in the shaping plan by most of the snakes. Targeting within their enclosure was learned within 4.2 sessions (SD = 1.47, range: 2–6), and targeting out of their enclosure was learned within 4 sessions (SD = 2.28, range: 2–8). Targeting to the shift container took the most amount of sessions for the snakes to learn; this behavior took an average of 11.8 sessions (SD = 2.56, range: 9–15) before progressing to the next step. The final behavior in the training plan, returning to their enclosure, required less sessions than the prior step. This behavior was learned within 5.5 sessions (SD = 2.34, range: 3–9). Every snake in clutch #4 performed the goal behavior at least once by the end of the study.

## 4. Discussion

In this study, a total of 23 *H. gigas* were trained using positive reinforcement methods. The goal behavior was to follow their target into a shift container and back to their primary habitat. Due to time constraints, a limited amount of data exists for the first three clutches. Despite the small amount of time with these clutches, each group of snakes moved beyond the classical conditioning phase of training and were interacting with the target in some way to earn a food reinforcer. Within 51 sessions, all individuals from the youngest clutch had fully learned and performed the goal behavior. During the training process, each snake was successfully weighed using the shifting behavior. The most common yellow zone behavior across all clutches was hooding. There was a difference in the most common red zone behaviors between the clutches. The youngest and oldest clutches most often hid as result of severe stress, whereas clutch #3 exhibited body flattening and clutch #2 attempted to bite the trainer most often. Based on limited data and anecdotal observations, it is possible clutch #2 adopted a proactive response to stress, while the others displayed a reactive response. Proactive responses are generally described as active behaviors such as biting or fleeing from the stressor; in contrast, reactive individuals are more likely to display passive behaviors including freezing or hiding [30].

The three clutches with previous aversive experiences during pre-study training sessions may serve as representation for the many snakes in human care that positive reinforcement and least invasive methods have yet to be used for. As snakes are long-lived animals and positive reinforcement training is a relatively new concept for the herpetoculture community, a significant number of captive snakes have likely had aversive experiences, ranging in severity. While the data for these three clutches is limited, it does indicate that there may be a difference in how these snakes responded to the training when aversive experiences occurred during sessions in comparison to the clutch that had no prior learning history.

For older snakes that have minimal to no training history or have experienced an aversive event, modifications to a training plan may be necessary to minimize stress and ensure keeper/animal safety. Some examples of these modifications include switching to a target that is a different shape and color if a previous target becomes associated with an aversive event, leaving a target in an enclosure for the snake to passively habituate to if the snake is shy or fearful of human presence, initiating a training session only when the animal is not hiding, taking breaks during sessions, and emphasizing the animal’s choice to participate in training sessions [25]. Research supports that there are physiological and behavioral detriments when personal control is absent and that the removal of choice, in and of itself, can be distressing [13]. Simply giving an animal the choice to participate is empowering and can mitigate fearful behaviors [20]. It is important to note that these modifications can also be beneficial for snakes that have not experienced aversive events.

There were several notable observations and challenges that arose during the training process. During early sessions, most snakes would not emerge from their hides, or they moved very slowly in the presence of the target. Presenting the target just outside of the containment area while holding the food reinforcer in the snakes’ enclosures appeared to be a sufficient solution, as the snakes become more comfortable with presence of the trainer and the target. For the very shy or fearful snakes, especially the ones with previous aversive experiences, the modifications previously mentioned were implemented.

Once the operant conditioning part of the process began and snakes were asked to approach or follow the target, striking and biting at the target often occurred and persisted throughout the rest of the process. It was concluded that this behavior was not a response to being fearful of the target, as snakes had ample space to retreat from the target and were given the opportunity to opt out of training sessions. The striking/biting is likely a species-typical response, given the natural feeding behaviors of wild *H. gigas*. Multiple strategies were used, depending on the individual, to minimize the occurrence of this behavior, including reinforcing before snakes reached the target, reinforcing the first tongue flick after striking, and presenting the target further away from snake initially. This behavior persisted in some learners for the duration of this study; however, it is likely that with additional sessions extinction of the striking/biting could occur.

The greatest challenge that arose, specifically with the youngest clutch, was a lack of focus from learners. This mainly occurred midway through the end of the training sessions. The main strategy for this was to offer snakes additional time in the training environments outside of normal sessions, which gave the snakes an opportunity for physical exercise and exposure to novel stimuli. However, this clutch still seemed distracted and appeared to find exploration more reinforcing than the food reward during training sessions, especially in later sessions. One factor that may have contributed to this was their style of housing. These snakes were housed in a rack throughout the duration of the study and did not receive the same amount of visual enrichment and physical space within their primary enclosure as they did in the training environments.

It has been previously documented that housing conditions of snakes can affect their cognitive abilities. Almli and Burghardt [31] found that rat snakes (*Elaphe obsolete*) housed in enriched conditions were less reactive and completed a problem-solving task faster than their counterparts housed in standard conditions. In another study, Nagabaskaran et al. [32] found that corn snakes (*Pantherophis guttata*) housed in enriched conditions could differentiate between familiar and unfamiliar human scents. Additionally, there is evidence that snakes will choose to spend more time in an enriched enclosure when given the choice to do so. Hoehfurtner et al. [33], found that most corn snakes in their study chose to spend the most time in an enriched enclosure, as opposed to one furnished with a hide, water bowl, and newspaper, after being exposed to both types of housing conditions. Behavioral diversity also appears to be affected by housing conditions. A study using royal pythons (*Python regius*) that were housed in both a rack system and enriched enclosures showed that when housed in enriched conditions, the snakes displayed significantly more species-appropriate behaviors [34]. Based on multiple species used in these studies, it is plausible to consider that the housing conditions for the *H. gigas* in the present study may have affected their behavior and learning during the process.

Targeting is an important behavior for captive animal management, but it is also serves as a powerful communication tool. In this study, targeting was used as a foundation for building the shifting behavior; however, it also informed snakes that food would be available and that a feeding response is an appropriate behavior in that context. Due to the false water cobra’s naturally intense feeding response and attraction to movement, providing a clear signal for this behavior is imperative to keeper safety within training sessions and during routine husbandry interactions. The target also functions as an agent of predictability, as it is something that the snakes are familiar with and produces a repeated positive outcome. Predictability has been shown to play a significant role in the wellbeing of captive animals, particularly in the presence of aversive stimuli [35]. While caretakers should strive to offer animals choices and minimize aversive experiences, it is not always possible to do so in a captive environment. Previous studies using rats, fish, and birds, have demonstrated that in the presence of an unavoidable aversive stimulus, animals prefer to receive a reliable signal prior to the event [35]. While not empirically evaluated, anecdotal reports indicate that this is likely true for snakes as well. Professional animal trainers recommend the use of a specific cue for snakes that indicates when choice is not an option [36].

Training provides a positive experience for animals, keepers, and for visitors to zoological facilities. Education and conservation are important facets of the mission statements of zoos. Animals can be trained to perform behaviors that showcase their natural, unique behaviors, as well as behaviors that reinforce conservation messaging. Whitehouse-Tedd, Spooner, and Whitehouse-Tedd [37] describe several case studies where animals were trained to perform different behaviors relevant to specific educational messaging, including free flight owls displaying flight and hunting behaviors, servals trained to jump to reach a target in a presentation about adaptations to a savannah ecosystem, and a penguin trained to throw away litter for a program that encourages visitors to dispose of trash in an appropriate manner. These types of encounters appear to be effective, especially when natural behaviors are displayed, with visitors being able to recall more specific information about the animals presented and the development of emotional bonds to specific animals, which is important for progressing conservation actions [38]. Educational encounters also play an important role in building empathy for less charismatic species. Knudson [39] found that touch tanks at aquariums influence empathy towards aquatic invertebrates.

While training was not the focus of these encounters, it does show that some interaction with less charismatic species can change visitors’ attitudes towards them. Snakes and other reptiles tend to fall within the category of less charismatic species, yet they are commonly used in educational programming. Displaying snakes in a manner that showcases their natural abilities and utilizing trained behaviors that place emphasis on their cognitive abilities may promote more empathetic attitudes towards them.

## 5. Conclusions

The use of positive reinforcement training for juvenile *H. gigas* resulted in a minimally invasive, choice-based method of moving snakes between their primary enclosures and a secondary shift container. All individuals in clutch #4 displayed full completion of the goal behavior, as well as a significant decrease in severe stress behaviors, indicating that this plan is a feasible and effective method for shaping behaviors in snakes with no prior training history. The responses displayed by the clutches with prior aversive experiences indicate that while the training is still possible, learning the behavior may occur at a slower rate and special accommodations may need to be made to avoid further behavioral fallout. The targeting and shifting behaviors trained in this study are foundational skills for more advanced behaviors that can be trained in the future. Positive reinforcement training is a powerful tool that can be used across taxa to produce a myriad of benefits for both keepers and animals, including giving animals choice and control over routine procedures, reducing stress, improving safety, and enhancing animal wellbeing.

## Figures and Tables

**Figure 1 animals-12-01229-f001:**
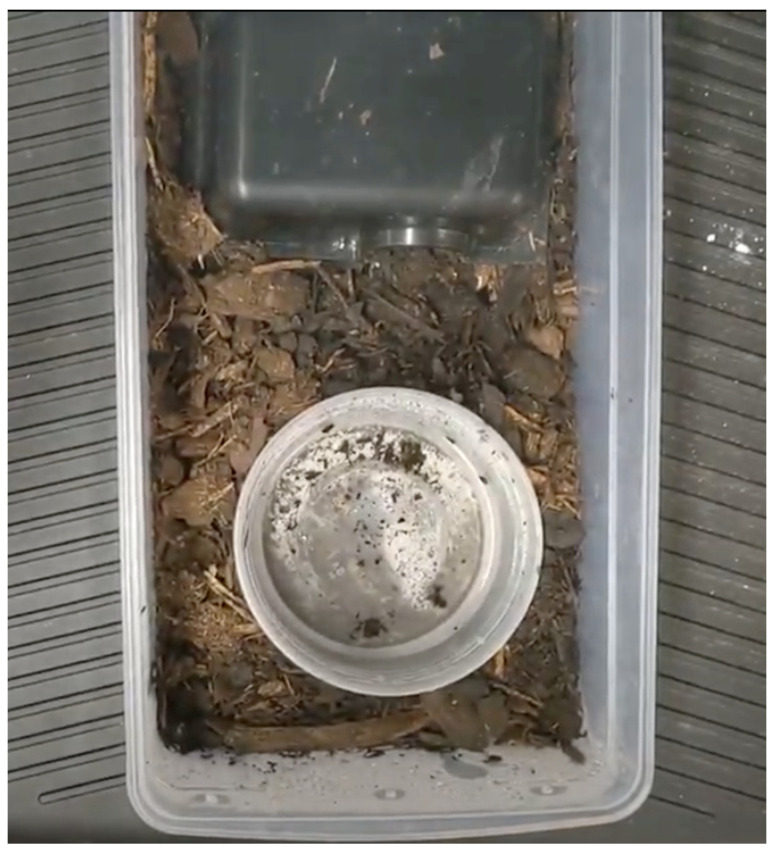
The initial housing setup for all snakes in the study is displayed. All enclosures contained a substrate mixture, a water bowl large enough for soaking, and a hide.

**Figure 2 animals-12-01229-f002:**
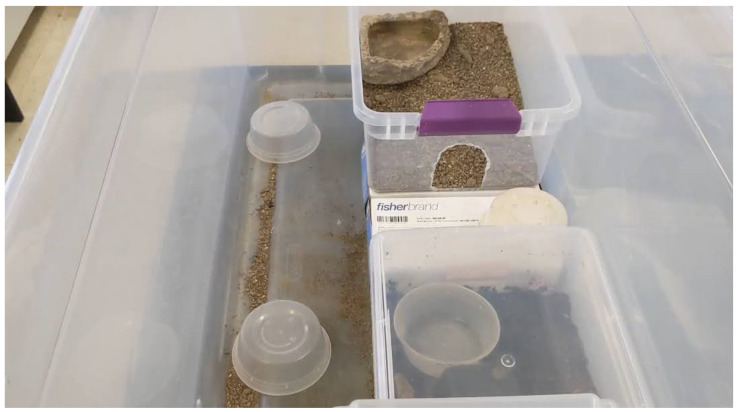
The initial training environment is displayed here. The training environment contains the first shift tub, three deli cups, and a snake’s primary enclosure. Glove boxes were used to make the entrance of the shift tub approximately the same height as the primary enclosure. The shift tub contains a substrate mixture and water bowl.

**Figure 3 animals-12-01229-f003:**
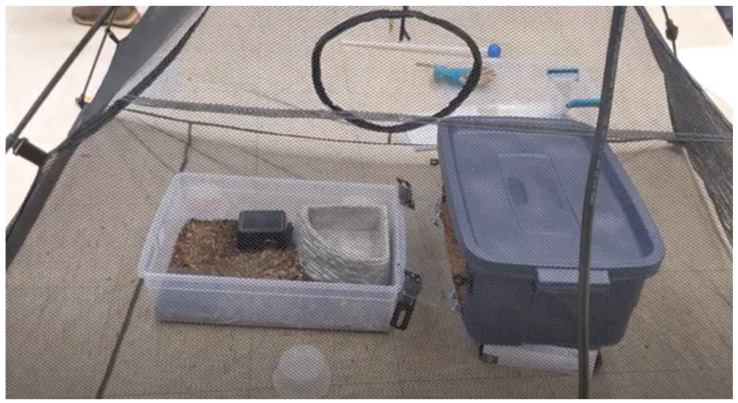
The larger training environment within a small tent. The tent training environment contains deli cups, a snake’s primary enclosure, and the larger shift tub. Glove boxes were used to elevate the shift tub so that the door was the same height as the primary enclosure. The shift tub contains the same substrate mixture as the first shift tub and a larger water bowl.

**Figure 4 animals-12-01229-f004:**
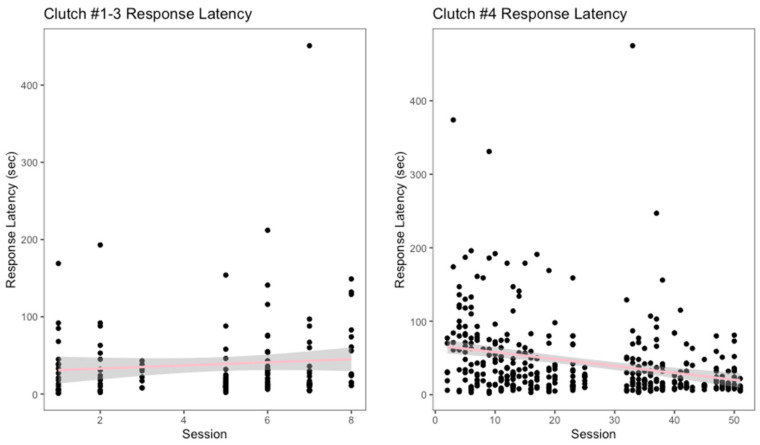
Response latency, measured in seconds, as the training sessions progressed is displayed above.

**Figure 5 animals-12-01229-f005:**
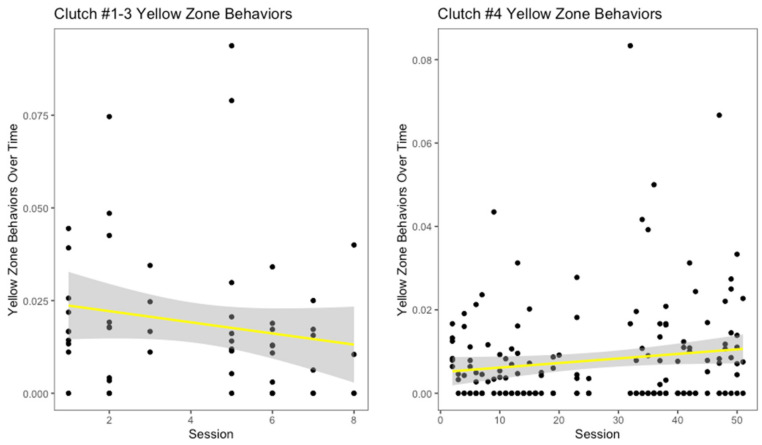
Moderate stress behaviors, also described as yellow zone behaviors, over time is displayed above. Freezing, hiding head, hooding, and nose rubbing were all considered yellow zone behaviors for *H. gigas*.

**Figure 6 animals-12-01229-f006:**
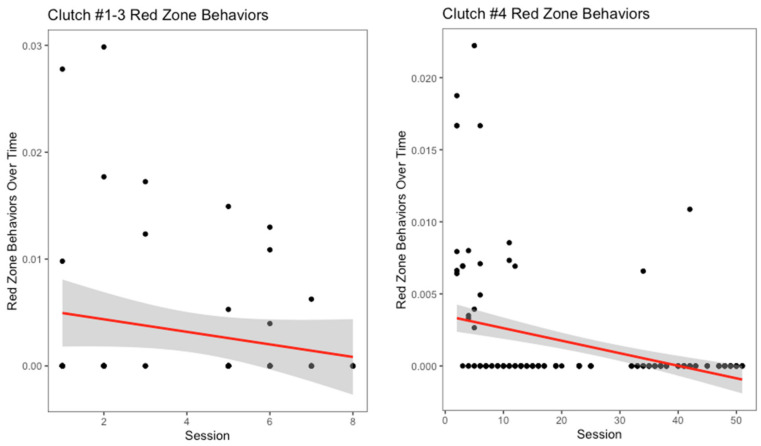
Severe stress behaviors, referred to as red zone behaviors, over time is displayed above. Biting, escaping from tub, hiding, body flattening, moving away, tail whipping, and defensive striking were all considered red zone behaviors for this species.

**Table 1 animals-12-01229-t001:** The clutch number, hatch date, and age at the beginning of the training are specified for the individuals used in this study. Clutch #1 and #4 had the same parents, while clutch #2 and #3 had unique parent pairings. Seven individuals were randomly selected from each clutch prior to the beginning of training.

Clutch	Hatch Date	Age at Start of Training	Number of Individuals Trained
#1	6/10/20	5 months	4 (3 removed)
#2	6/28/20	5 months	7
#3	7/13/20	4 months	6 (1 removed)
#4	8/21/20	3 months	6 (1 removed)

**Table 2 animals-12-01229-t002:** Stress behaviors observed throughout the training process. Yellow zone (Yellow rows) are indicative of moderate stress levels, and red zone (Red rows) behaviors describe severe stress levels.

Behavior	Description
Freezing	No movement or tongue flicking, does not respond to stimuli
Hiding head	Attempting to put head under something
Hooding	Flattening upper 1/3 of body
Nose rubbing	Rubbing nose on tub/object surface
Biting	Attempts to bite trainer
Escaping	At least 1/3 of body out of tub, accompanied by other yellow/red zone behaviors
Hiding	Retreating to hide and remaining fully in hide, preceded by other yellow/red zone behaviors
Body flattening	Flattening upper 2/3 of body or entire body
Moving away	Fleeing from target, food, or trainer within tub
Tail whipping	Tail whipping, accompanied by other yellow/red zone behaviors
Defensive striking	Striking at target when too closely, accompanied by other yellow/red zone behaviors

**Table 3 animals-12-01229-t003:** The number of training sessions for each snake in the fourth clutch to confidently perform the subsequent step. The subsequent step for the final training step listed below was completing the full behavior of entering the shift container and returning to their enclosure. False water cobra is abbreviated as FWC.

	Training Progression					
Training Steps	FWC 8	FWC 9	FWC 10	FWC 11	FWC 12	FWC 14
Classical conditioning	9	11	7	10	9	9
Targeting in enclosure	4	3	6	2	5	5
Targeting out of enclosure	2	5	2	3	8	4
Targeting to shift container	15	11	15	10	11	9
Returning to enclosure	6	5	9	3	7	3

## Data Availability

Data is available from the primary author upon request.

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
