# Peer review of "Using Classical and Operant Conditioning to Train a Shifting Behavior in Juvenile False Water Cobras (Hydrodynastes gigas)"

_animals, 2022, doi:10.3390/ani12101229_

Round 1

Reviewer 1 Report

Reptile training and snake training is an under-focused topic. While this is a good paper, here are two citations (not peer reviewed- but extremely relevant) that were not included. 

Gerrtis, J. and L. Augustine. 2013.Multiple snakes, multiple problems. ABMA Wellsprings, 13(1): 9-10.

Gerrtis, J. and L. Augustine. 2013. Multiple snakes, multiple problems. AAZK Animal Keeper’s Forum, 40(7): 318-319.

I think a bigger focus on the housing conditions both before and during the study is needed. Temperature is an important factor. Incubation is also relevant- incubated apart in an incubator or as a clutch in an incubator or as a clutch by the dam?  could have behavioral implications and may correlate with difference seen between clutches. Same goes for hatchling husbandry and care.

photos of enclosures?

consistent trainer for each snake and each trial?

any ability to count rate of tongue flick as a metric?

The more details on these factors can help identify correlations and future directions of study

Reviewer 2 Report

The present study tests ability of snakes to learn specific behaviors by training with positive reinforcement. Specifically designed training could drastically reduce stress of animals in captivity and  improve their well being. Results of this and similar studies, which combines known life-history traits and specific animal behaviors, are very important for animal welfare. The idea is well explained. However, the study design is not explained in enough details. Materials and methods part needs to be rewritten. That would help to understand results and their implications and would be helpful in reading discussion. Are there any ethical issues, were some permissions obtained before start of the experiment? Who did trainings, was it the same person or different persons were present during training seasons? Were juveniles trained throughout ontogeny or at specific points in time i.e. how old were they? This is partly explained on graphs in results part but should be clearly indicated in MM part. Does different approaches to training for different animals impact their response in further training? It is not clear enough why groups for statistical analyses were divided the way they are. For example, difference in age between clutches #1 and #3 is similar to difference in age of clutches #3 and #4. Please find specific comments below.

Line 182: Does this differences in plan have an impact on results?

Lines 183-185: Was the same time of day each day and for all subjects?

Minor comments

Line 59: add space between “Savastano et al. ” and “ [4] ”

Line 74: add space between “[8].” and “Training”

Line 104:  suggestion: remove Emer et al. and leave just reference number, it is easier to follow or reformulate sentence

Line 110: add full stop after “[19]”

Line 117: P is in bold

Line 120: What do you mean by “natural history information”, maybe life-history traits?

Line 141: change “H. gigas” into “Hydrodynastes gigas” because it is after full stop

Line 151: put a reference for Torrini method

Line 178: replace “natural history” with “life history traits”

Line 235: isn’t that Table 2?

Line 241: Table 2 – replace “def.” in last row with full word

Line 245and 247: “older” instead of “oldest”?

Line 288: H. gigans should be in italic
